# Maternal and neonatal outcomes of antihypertensive treatment in pregnancy: A retrospective cohort study

Sascha Dublin[1,2]*, Abisola Idu[1], Lyndsay A. Avalos[3], T. Craig Cheetham[4], Thomas R. Easterling[5], Lu Chen[1¤a], Victoria L. Holt[2], Nerissa Nance[3], Zoe Bider-Canfield[6¤b], Romain S. Neugebauer[3], Kristi Reynolds[6], Sylvia E. Badon[3], Susan M. Shortreed[1,7]

**1** Kaiser Permanente Washington Health Research Institute, Kaiser Permanente Washington, Seattle, Washington, United States of America, **2** Department of Epidemiology, University of Washington, Seattle, Washington, United States of America, **3** Division of Research, Kaiser Permanente Northern California, Oakland, California, United States of America, **4** School of Pharmacy, Chapman University, Irvine, California, United States of America, **5** Department of Obstetrics & Gynecology, University of Washington, Seattle, Washington, United States of America, **6** Department of Research and Evaluation, Kaiser Permanente Southern California, Pasadena, California, United States of America, **7** Department of Biostatistics, University of Washington, Seattle, Washington, United States of America

¤a Current address: PHC Data, Analytic and Imaging, Genentech (a member of Roche Group), South San Francisco, California, United States of America
¤b Current address: Evidence Generation and Strategic Collaborations, Medical Affairs, Hoffman-La Roche, Mississauga, Ontario, Canada
* Sascha.Dublin@kp.org

**Data Availability Statement:** Project data come from patient electronic health records and birth certificates from the states of California and Washington. Data from Kaiser Permanente

## Abstract

### Objective

To compare maternal and infant outcomes with different antihypertensive medications in pregnancy.

### Design

Retrospective cohort study.

### Setting

Kaiser Permanente, a large healthcare system in the United States.

### Population

Women aged 15–49 years with a singleton birth from 2005–2014 treated for hypertension.

### Methods

We identified medication exposure from automated pharmacy data based on the earliest dispensing after the first prenatal visit. Using logistic regression, we calculated weighted outcome prevalences, adjusted odds ratios (aORs) and 95% confidence intervals, with inverse probability of treatment weighting to address confounding.

electronic health records are proprietary to Kaiser Permanente and cannot be shared without confidentiality agreements. Data from electronic health records and state birth certificates cannot be publicly disclosed due to concerns about sensitive patient/personal data. Requests for data may be sent to the MOHIP Data Access Committee, which consists of Drs. Dublin (Sascha.Dublin@kp.org) and Shortreed (Susan.M.Shortreed@kp.org) from Kaiser Permanente Washington, Dr. Avalos (Lyndsay.A.Avalos@kp.org from Kaiser Permanente Northern California, and Dr. Reynolds (Kristi.Reynolds@kp.org) from Kaiser Permanente Southern California. Requests may also be sent to the Kaiser Permanente Washington Vice President for Research and Health Care Innovation, Dr. Rita Mangione-Smith, who will facilitate the process (Rita.M.Mangione-Smith@kp.org). Researchers wishing to obtain birth certificate data from Washington State may contact the Washington State IRB at wsirb@dshs.wa.gov or visit their website (https://www.dshs.wa.gov/ffa/human-research-review-section) to learn more. Researchers wishing to obtain California birth certificate data can find information about the California Committee for the Protection of Human Subjects at their website, https://www.chhs.ca.gov/cphs/, or email cphs@chhs.ca.gov. IRB approval is required, but not sufficient, to access study data. Requests will be reviewed by the study Data Access Committee for scientific merit, human subjects considerations, and corporate legal obligations. After approval, a data sharing agreement will be created, approved, and signed.

**Funding:** SD received grant R01HD082141 from the National Institute on Child Health and Human Development, https://www.nichd.nih.gov/ LC was funded as a postdoctoral fellow by the Group Health Foundation (no grant number); this incarnation of the foundation no longer exists. Group Health is the former name of the healthcare system now known as Kaiser Permanente Washington. When Kaiser Permanente purchased Group Health, they placed money into a new foundation that is also called the Group Health Foundation. It has a URL but is not the entity that funded Dr. Chen. The funders had no role in the study design, data collection and analysis, decision to publish, or preparation of the manuscript.

**Competing interests:** I have read the journal's policy and the authors of this manuscript have the following competing interests: SD received a grant to support this work from the National Institute on Child Health and Human Development. She has also received grant support from GSK for unrelated work. LC received a postdoctoral fellowship from

## Main outcome measures

Small for gestational age, preterm delivery, neonatal and maternal intensive care unit (ICU) admission, preeclampsia, and stillbirth or termination at > 20 weeks.

## Results

Among 6346 deliveries, 87% with chronic hypertension, the risk of the infant being small for gestational age (birthweight < 10th percentile) was lower with methyldopa than labetalol (prevalence 13.6% vs. 16.6%; aOR 0.77, 95% CI 0.63 to 0.92). For birthweight < 3rd percentile the aOR was 0.57 (0.39 to 0.80). Compared with labetalol (26.0%), risk of preterm delivery was similar for methyldopa (26.5%; aOR 1.10 [0.95 to 1.27]) and slightly higher for nifedipine (28.5%; aOR 1.25 [1.06 to 1.46]) and other β-blockers (31.2%; aOR 1.58 [1.07 to 2.23]). Neonatal ICU admission was more common with nifedipine than labetalol (25.9% vs. 23.3%, aOR 1.21 [1.02 to 1.43]) but not elevated with methyldopa. Risks of other outcomes did not differ by medication.

## Conclusions

Risk of most outcomes was similar comparing labetalol, methyldopa and nifedipine. Risk of the infant being small for gestational age was substantially lower for methyldopa, suggesting this medication may warrant further consideration.

## Introduction

Hypertensive disorders affect 5–10% of pregnancies [1], increasing the risk of fetal growth restriction, stillbirth and other adverse outcomes [2–5]. About 160,000 pregnant women take antihypertensive medications annually in the US [2], yet it is unclear which medication results in the best outcomes for women and infants. Current US and UK guidelines recommend labetalol and nifedipine over methyldopa, while acknowledging uncertainty [6, 7]. The International Society for the Study of Hypertension in Pregnancy has stated that both methyldopa and nifedipine are acceptable [8].

Randomized clinical trials (RCTs) have not provided definitive evidence because they have had small sample sizes and heterogeneous methods. If sufficient data were available from RCTs, a meta-analysis could be performed to compare outcomes with different medications, but unfortunately data are sparse. A 2018 Cochrane meta-analysis [9] identified 29 RCTs that compared antihypertensive medications head-to-head; taken together, these trials included a total of only 2774 women. The trials were heterogeneous, examining many different medications, which resulted in very small sample sizes for specific comparisons. The only definitive finding from the meta-analysis was that β-blockers and calcium channel blockers appeared more effective than methyldopa at preventing severe hypertension [9]. For other outcomes, there were no statistically significant differences, which is understandable because often only a few trials were included, leading to low precision and wide confidence intervals. The Cochrane meta-analysis grouped together all β-blockers, which may obscure important differences between individual medications, especially as labetalol has different receptor specificity than other commonly used β-blockers. A recent RCT reported that methyldopa was associated with significantly lower risk of small for gestational age (SGA) and NICU admission compared to

the Group Health Foundation. She is now employed by Genentech (a member of Roche Group). ZBC is now employed by Roche Pharmaceuticals. TRE has consulted for Alnylam Pharmaceuticals, DiabetOmics, and Ferring Pharmaceuticals. SMS has received grant funding through her institutions from Syneos Health for work unrelated to this study. LAA received funding through her institution from Bausch Health Companies and KR from Amgen, Novartis and Merck & Co, all for work unrelated to this study. This does not alter our adherence to PLOS ONE policies on sharing data and materials.

labetalol, with odds ratios on the order of 0.40, and that the two medications were associated with similar risk of severe maternal hypertension or preeclampsia [10]. However, the sample size was small (~150 women per arm) and many of their risk estimates had wide confidence intervals. Due to the limitations outlined above, existing RCT data are not adequate to guide choice of medications for the treatment of hypertension in pregnancy.

When RCT data are insufficient, as is often the case for medication use in pregnancy, rigorous observational studies can provide useful information. One observational analysis [11] used data from the Control of Hypertension in Pregnancy Study [12], which randomized pregnant women to tight vs. less tight blood pressure control but did not dictate which medications were used. The post hoc observational analysis compared pregnancy outcomes with methyldopa vs. labetalol (an observational comparison, since choice of medications was not randomized) and found better outcomes with methyldopa [11]. Other antihypertensive medications were not examined. Several other observational studies have been conducted, but they had important methodologic limitations which make it difficult to draw causal inferences. Many of these studies compared women treated with an antihypertensive medication to unexposed women from the general pregnant population [3, 5, 13–16], most of whom presumably did not have hypertension. Since hypertension increases the risk of adverse pregnancy outcomes, these studies are vulnerable to confounding by indication and cannot shed light on the risks of treatment vs. those due to hypertension. Additional studies are needed using rigorous methods that can support causal inference.

Because additional evidence is needed, we sought to compare the risk of important maternal and infant outcomes with use of different antihypertensive medications using electronic health records (EHR) data for a large, diverse US population.

## Methods

### Overview

This retrospective cohort study was conducted at Kaiser Permanente, a US healthcare system providing health care and insurance coverage. Participating regions were Washington, Southern California, and Northern California, which together serve about 8 million people generally representative of the surrounding communities [17]. Data came from EHRs and linked birth certificate data. These data have been used in many pregnancy studies [18–21], and important variables and methods have been validated [22–25]. The study used rigorous causal inference methods [26, 27], including following recommended principles for emulating a target trial [28], using active comparators (comparing outcomes with one medication vs. another used for the same indication) [29], and addressing confounding using inverse probability weighting [27]. Study procedures were approved by the regions' institutional review boards and those of Washington State and California, with a waiver of consent.

### Study population

The population was women age 15–49 years with a singleton live or stillbirth from 2005 through 2014. Women were required to be enrolled in Kaiser Permanente from 16 weeks' gestation through delivery, to have at least one blood pressure (BP) measured before 20 weeks, and to have chronic or gestational hypertension (defined from BP values, diagnosis codes and medication fills; our algorithm is shown in S1 Table in S1 File and has been published [30]). We included both chronic or gestational hypertension because in clinical practice, it can be difficult to determine which type of hypertension is present and because these conditions may represent different points on a continuum of disease.

Women had to have filled at least one antihypertensive medication before 36 weeks gestation, to be on monotherapy, and to have been enrolled in Kaiser Permanente for at least 150 days before their qualifying fill. They could contribute more than one pregnancy to these analyses. We excluded deliveries exposed to teratogenic medications or certain high-risk maternal medical conditions (see S1 Table in S1 File for more information). The sample size was determined by the number of eligible births.

## Exposures

From computerized pharmacy data, we identified fills for labetalol, methyldopa, nifedipine and other β-blockers (S1 Table in S1 File). These data are recorded prospectively when medications are dispensed, eliminating the biases that can arise in some retrospective observational studies (for instance, studies that interview women after delivery about medications taken in pregnancy.) Unlike many prior studies, we considered labetalol separately from other β-blockers because it is a combined α and β-blocker and unlike other β-blockers, it is recommended as first-line in US guidelines [6]. Exposure was defined based on the earliest fill after the first prenatal visit (typically at 8–10 weeks' gestation) or, if the visit date was unknown, at $\geq 10$ weeks gestation; we called this the 'index fill'. Using intent to treat principles, women's exposure status was fixed rather than time-varying, because subsequent medication switches could be affected by the initial medication choice.

## Outcomes

Outcomes included SGA, preterm delivery, neonatal intensive care unit (NICU) admission, preeclampsia, maternal ICU admission, and stillbirth or termination at > 20 weeks. SGA was defined using sex and race-specific US birthweight curves [31]. The primary outcome was birthweight $<10^{th}$ percentile for gestational age and a secondary outcome $< 3^{rd}$ percentile. Deliveries missing birthweight (n = 32) were excluded from SGA analyses. We defined preterm delivery using gestational age from the EHR (preferentially) or birth certificate data, with the primary outcome being delivery before 37 weeks gestation and a secondary outcome, delivery before 34 weeks. We considered preterm delivery a potential measure of medication effectiveness, because less effective medications could lead to higher risk of uncontrolled maternal hypertension or fetal growth restriction (a potential consequence of severe hypertension) and via these pathways, to indicated preterm delivery. The automated data available to us do not reliably distinguish spontaneous vs. indicated preterm births. We identified ICU admissions using EHR data. Preeclampsia was identified from inpatient diagnosis codes after 20 weeks' gestation, an approach with a positive predictive value of 90% [32]. We reviewed 45 charts meeting those criteria and found a positive predictive value of 93%. We identified preeclampsia cases with "severe features" using modified criteria from the American College of Obstetricians & Gynecologists [33], drawing on BP values, laboratory results and diagnosis codes (S1 Table in S1 File).

Potential stillbirths and terminations after 20 weeks' gestation were identified using EHR data; we included as outcomes the 76% of potential cases validated through medical record review or linkage to fetal death certificates. We grouped together stillbirths and terminations for several reasons. Terminations after 20 weeks might be done for fetal anomalies, which could in theory be affected by medication choice, as there is no definitive evidence about birth defect risk for some widely used antihypertensive medications. Also, the decision to terminate might be influenced by severe uncontrolled maternal hypertension, which could be a consequence of the initial medication choice. Finally, we hypothesized that variation in coding

might lead to similar clinical scenarios being classified as either a stillbirth or termination in different instances.

## Covariates

Covariates included maternal age at delivery, Kaiser Permanente region, delivery year, hypertension type (chronic or gestational), BP values, race/ethnicity, parity, maternal education, pregestational diabetes, depression, tobacco use, body mass index (BMI), and prior use of certain medications (S1 Table in S1 File). Hypertension was categorized as chronic if it was present prior to pregnancy or during the first 20 weeks gestation and as gestational otherwise. To account for hypertension severity, we identified the most recent BP value prior to the index fill and also determined whether a woman experienced one or more BPs $\geq$ 160/110 before pregnancy or during this pregnancy before the index fill. We categorized history of antihypertensive medication use as no use prior to the index fill, continuous use up to the index fill (allowing for 80% adherence), or prior use with a gap. Other covariates included prior use of angiotensin converting enzyme inhibitors, angiotensin receptor blockers, thiazide diuretics, diabetes medications, benzodiazepines, statins, antidepressants or antiseizure medications.

## Statistical analyses

Descriptive analyses included counts and proportions for categorical variables and means and standard deviations for continuous variables. Primary analyses used logistic regression to model study outcomes, with labetalol as the referent group. Inverse probability of treatment weights (IPTW) were used to account for confounding. We calculated weighted outcome prevalences for each medication group and adjusted odds ratios (aORs) and 95% confidence intervals (CIs). We used the bootstrap to account for multiple pregnancies per woman and for the estimation of the weights [34, 35]. Treatment weights were generated from propensity scores calculated using a multinomial logistic regression model including all covariates shown in Table 1 except for BMI, education, parity, and timing of prenatal care. We omitted these variables because they were well balanced before weighting and a small proportion of deliveries had missing information for each of these characteristics. S2 Table in S1 File lists variables in the propensity score. To improve statistical efficiency, we calculated stabilized weights including some baseline covariates in both the outcome model and the numerator of the weights [36, 37]. These were Kaiser Permanente region, race/ethnicity, diabetes, type of hypertension (chronic vs. gestational), and gestational age at index fill.

For statistical modeling, we categorized delivery year as 2005–2008, 2009–2010, 2011–2012, and 2013–2014. We grouped together the four earliest years because very few deliveries were included from 2005–2006, when only one region had electronic BP values available. Maternal age was categorized as < 30, 30–34, 35–39 or $\geq$ 40 years. Gestational age at the index fill was modeled as a linear spline with knots at 140 and 210 days. The systolic and diastolic BP values closest to the index fill were modeled using linear splines, with knots at 140 mm Hg and 90 mm Hg respectively. Deliveries missing race/ethnicity (0.5%) were grouped with those with "other" race/ethnicity and treated as a category of race/ethnicity in statistical models.

To assess covariate balance, we calculated the average standardized mean differences across all treatment groups before and after IPTW [38, 39].

We excluded stillbirths/terminations from analyses of SGA, NICU and preterm delivery because they are competing events. We used inverse probability of censoring weights to account for possible bias due to excluding stillbirths; S3 Table in S1 File lists the variables used to model these weights.

**Table 1. Baseline characteristics of the population before weighting, by treatment group[a].**

| Characteristic (number (%) unless otherwise stated) | Labetalol N = 3017 | Methyldopa N = 1834 | Nifedipine N = 1105 | Other β-blockers N = 390 |
|---|---|---|---|---|
| Maternal age, yrs, mean±SD | 33.5±5.2 | 33.9±5.3 | 33.2±5.6 | 33.8±5.2 |
| Nulliparous[b] | 1109 (36.8) | 699 (38.1) | 422 (38.2) | 139 (35.6) |
| Race/ethnicity | | | | |
| White, non-Hispanic | 1011 (33.5) | 551 (30.0) | 369 (33.4) | 175 (44.9) |
| Hispanic | 928 (30.8) | 604 (32.9) | 275 (24.9) | 75 (19.2) |
| Black, non-Hispanic | 470 (15.6) | 242 (13.2) | 184 (16.7) | 60 (15.4) |
| Asian | 575 (19.1) | 414 (22.6) | 261 (23.6) | 73 (18.7) |
| Obese (BMI $\geq$ 30 kg/m$^2$)[b] | 1796 (59.5) | 968 (52.8) | 581 (52.6) | 223 (57.2) |
| Tobacco use | 150 (5.0) | 68 (3.7) | 53 (4.8) | 21 (5.4) |
| Chronic hypertension | 2550 (84.5) | 1693 (92.3) | 910 (82.4) | 360 (92.3) |
| Pre-gestational diabetes | 560 (18.6) | 355 (19.4) | 250 (22.6) | 68 (17.4) |
| Prenatal care in trimester 1[b] | 2653 (87.9) | 1402 (76.4) | 922 (83.4) | 277 (71.0) |
| Gestational age at index fill (weeks), mean±SD | 18.8±9.6 | 16.5±8.0 | 20.8±9.6 | 17.1±7.9 |
| Systolic BP, mm Hg; mean±SD[c] | 142.8 (17.2) | 138.7 (15.7) | 138.9 (17.0) | 132.8 (17.3) |
| Diastolic BP, mm Hg; mean±SD[c] | 88.0 (12.1) | 84.9 (10.9) | 84.9 (12.9) | 80.7 (11.9) |
| Prior antihypertensive medication use | | | | |
| Prior use, continuous | 1018 (33.7) | 813 (44.3) | 348 (31.5) | 162 (41.5) |
| Prior use with a gap | 839 (27.8) | 577 (31.5) | 296 (26.8) | 127 (32.6) |
| No prior use | 1160 (38.4) | 444 (24.2) | 461 (41.7) | 101 (25.9) |
| Delivery year | | | | |
| 2005–2008 | 500 (16.6) | 704 (38.4) | 314 (28.4) | 139 (35.6) |
| 2009–2010 | 713 (23.6) | 489 (26.7) | 250 (22.6) | 99 (25.4) |
| 2011–2012 | 860 (28.5) | 348 (19.0) | 287 (26.0) | 89 (22.8) |
| 2013–2014 | 944 (31.3) | 293 (16.0) | 254 (23.0) | 63 (16.2) |

Abbreviations: BMI, body mass index; BP, blood pressure; SD, standard deviation.

[a]All characteristics measured prior to the index medication fill, except for delivery year. The proportion with missing data across groups was as follows: for parity, 2.8 to 4.2%; for race/ethnicity, 0.4 to 1%; and for BMI, 3.7 to 13.9%. No pregnancies had missing data for other listed variables.

[b]Covariate not in the propensity score model.

[c]Most recent BP prior to index fill of antihypertensive medication.

In sensitivity analyses, we restricted the analysis to women with chronic hypertension (87% of the population) and excluded women with pregestational diabetes. In subgroup analyses, we examined new users separately from women with prior antihypertensive treatment. Analyses were performed using R, version 3.5.

## Funding

This study was funded by the US National Institute on Child Health and Human Development grant R01HD082141. The Group Health Foundation funded Dr. Chen's fellowship. The funders did not play a role in conducting the research or writing the paper.

## Results

Among 6346 eligible deliveries, there were 3017 (48%) where the woman had taken labetalol, 1834 (29%) methyldopa, 1105 (17%) nifedipine, and 390 (6%) other β-blockers. Fig 1 shows the impact of inclusion and exclusion criteria on the study population.

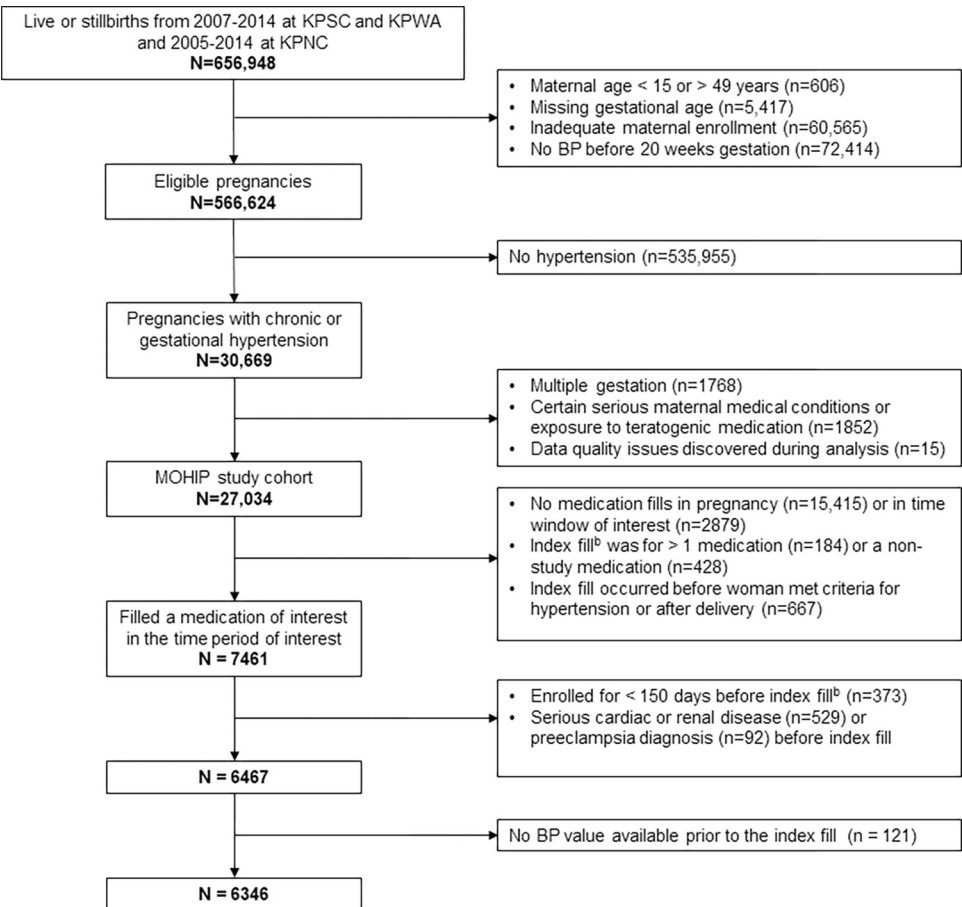

**Fig 1. Impact of inclusion and exclusion criteria on study population.** [a] Abbreviations: BP, blood pressure; KPNC, Kaiser Permanente Northern California; KPSC, Kaiser Permanente Southern California; KPWA, Kaiser Permanente Washington. [a]A woman may meet more than one exclusion criterion within a box. Detailed information about inclusion and exclusion criteria is found in S1 Table in S1 File. [b]The index fill was defined as the earliest fill after the first prenatal visit (typically at 8–10 weeks' gestation) or, if the visit date was not known, at $\geq$ 10 weeks gestation.

Mean maternal age was 33.6 years, 87% had chronic hypertension, and the mean gestational age at the index fill was 18.4 weeks. Many women (37%) were taking antihypertensive medication continuously prior to the index fill, and mean BPs prior to the index fill suggest that their hypertension was on average fairly well controlled. Table 1 shows baseline characteristics by treatment group, and S4 Table in S1 File provides more detailed information for an expanded list of baseline characteristics. S5 Table in S1 File shows characteristics by treatment group after IPTW and demonstrates that overall, these were well balanced (standardized mean difference < 0.1), except for those characteristics included in the outcome model, which are not expected to be balanced by IPTW. After IPTW, the group exposed to other β-blockers looked modestly different from the other groups, likely due to this group's small size. S6 Table and S1 Fig in S1 File describe the distributions of propensity scores and weights.

Most women did not switch medications after their index fill. The proportion of women who later filled a different medication was 15% overall, ranging from 11 to 22% for different exposure groups.

Table 2 provides crude counts of outcomes by treatment group. Fig 2 shows the risk of maternal and neonatal outcomes comparing different medications, with labetalol as the

**Table 2. Counts of maternal and neonatal outcomes by treatment group.**

| Outcome and Medication Class | Outcomes/Exposed Pregnancies[a] | | | |
|---|---|---|---|---|
| | **Labetalol** | **Methyldopa** | **Nifedipine** | **Other β-blockers** |
| Preeclampsia | 1001/3017 | 523/1834 | 338/1105 | 91/390 |
| Preeclampsia with severe features | 786/3017 | 351/1834 | 230/1105 | 56/390 |
| Maternal ICU | 61/3017 | 37/1834 | 20/1105 | 14/390 |
| Stillbirth or termination | 41/3017 | 18/1834 | 13/1105 | 2/390 |
| SGA < 10th percentile | 512/2962 | 231/1805 | 159/1091 | 66/382 |
| SGA < 3rd percentile | 145/2962 | 49/1805 | 41/1091 | 19/382 |
| Preterm delivery < 37 weeks | 812/2976 | 485/1816 | 352/1092 | 87/388 |
| Preterm delivery < 34 weeks | 299/2976 | 161/1816 | 112/1092 | 27/388 |
| Neonatal ICU admission | 726/2976 | 426/1816 | 296/1092 | 86/388 |

Abbreviations: OR, odds ratio; CI, confidence interval; SGA, small for gestational age; ICU, intensive care unit.

[a]Actual numbers prior to inverse probability of treatment weighting. The population for different outcomes differs slightly because pregnancy losses were not included in the denominator for SGA, preterm delivery, or neonatal ICU admission, and because 32 deliveries missing infant birthweight were excluded from analyses of SGA.

referent group. We present weighted prevalences for outcomes after accounting for confounders together with adjusted ORs and 95% CIs. For SGA < 10th percentile, risk was lower with methyldopa than labetalol (weighted prevalence 13.6% vs. 16.6%; aOR 0.77, 95% CI 0.63 to 0.92), and the association was stronger for birthweight < 3rd percentile (aOR 0.57, 95% CI 0.39 to 0.80). The mean birthweight after IPTW was 3002 ± 797 g for labetalol, 3060 ± 788 g for methyldopa, 3033 ± 798 g for nifedipine, and 2944 ± 791 g for other β-blockers.

Preterm delivery was slightly more common with nifedipine than labetalol (28.5% vs. 26.0%; aOR 1.25, 95% CI 1.06 to 1.46), as was NICU admission (25.9% vs. 23.3%; aOR 1.21, 95% CI 1.02 to 1.43). β-blockers other than labetalol were associated with higher risk of preterm delivery (aOR 1.58, 95% CI 1.07 to 2.23). Methyldopa and labetalol conveyed similar risks of preterm delivery and NICU admission. After IPTW, the mean gestational age at delivery was 37.6 ± 2.8 weeks for labetalol, 37.6 ± 2.8 weeks for methyldopa, 37.4 ± 2.8 weeks for nifedipine, and 37.4 ± 2.8 weeks for other β-blockers.

There was no significant association between medication type and risk of preeclampsia (overall or with severe features), maternal ICU admission, or stillbirth/termination.

Results of sensitivity and subgroup analyses are shown in S2-S5 Figs in S1 File. Results did not change when we restricted the population to women with chronic hypertension, who made up 87% of the population. Results also did not change when we excluded women with pregestational diabetes. Some findings appeared qualitatively different when we limited analyses to new users; in this group, there was a suggestion of lower risk for many outcomes with methyldopa than with labetalol, with aORs around 0.5 to 0.7 (though most were not statistically significant).

## Discussion

In this large retrospective cohort study, the prevalence of many maternal and neonatal outcomes was similar with use of different antihypertensive medications. Compared to labetalol, the risk of SGA was significantly lower with methyldopa.

Other studies have examined outcomes with use of different antihypertensive medications. Most prior studies were small, yielding inconclusive results, and many observational studies compared treated women to healthy pregnant women, making confounding likely. Our finding of lower SGA risk with methyldopa compared to labetalol (aOR 0.77, 95% CI 0.63 to 0.92)

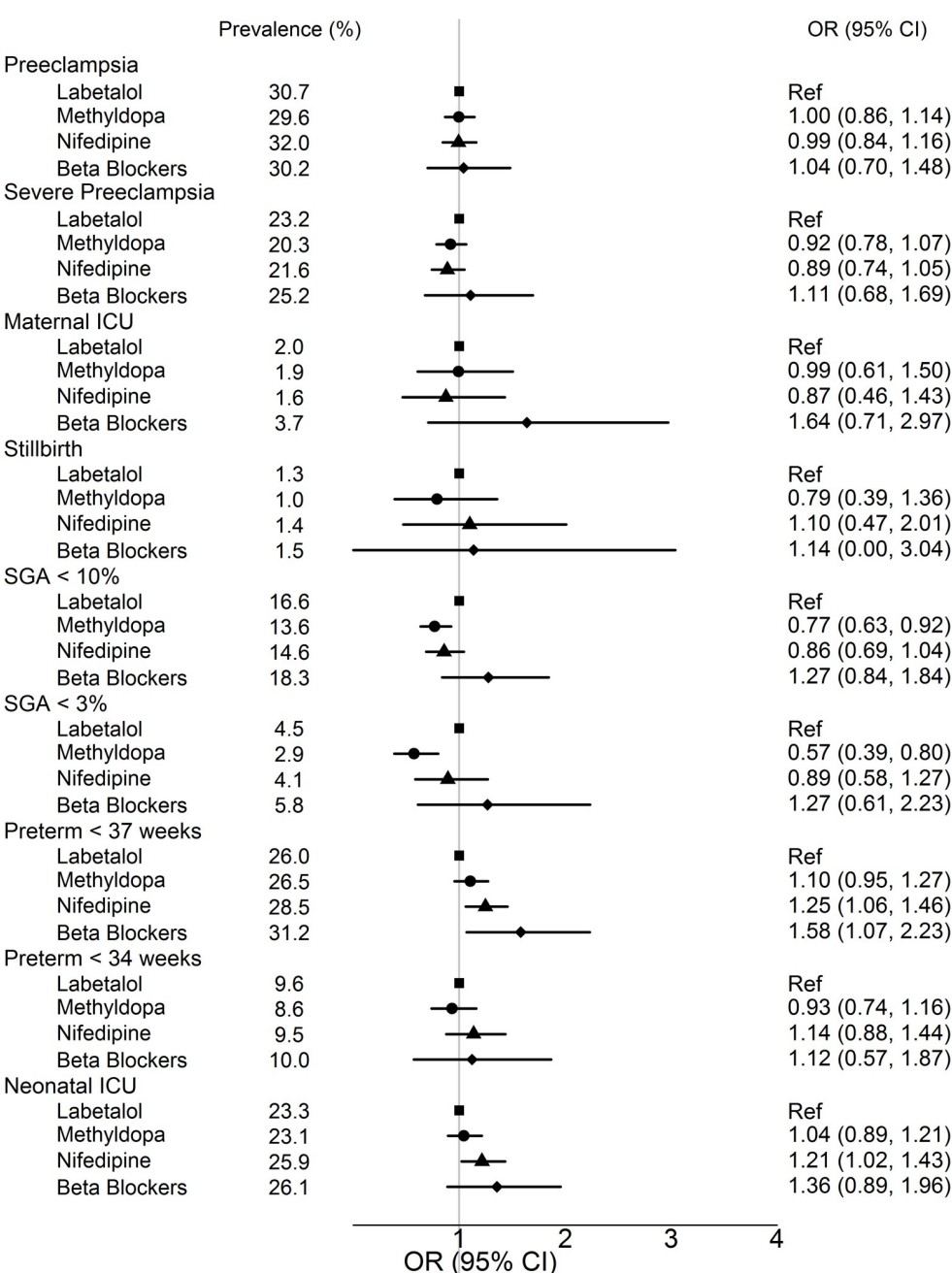

**Fig 2. Risk of maternal and neonatal outcomes with use of different antihypertensive medications in pregnancy\*.**
Abbreviations: OR, odds ratio; CI, confidence interval; SGA, small for gestational age; ICU, intensive care unit. *ORs and 95% CIs are calculated after inverse probability of treatment weighting. Labetalol is the referent group. The population for different outcomes differs slightly because pregnancy losses were not included in the denominator for SGA, preterm delivery, or neonatal ICU admission. For most outcomes, the total N is 6346, for SGA the total N is 6240, and for preterm delivery and NICU admission the total N is 6272. **Weighted prevalence in the subgroup, calculated using inverse probability of treatment weighting with unstabilized weights.

is consistent with one recent RCT, which found the prevalence of SGA was much lower with methyldopa than labetalol (21% vs. 41%;OR 0.37, 95% CI 0.23–0.61) [10]. Similar results were found by Magee et al. in a secondary analysis of RCT data [11]. The Cochrane meta-analysis of

RCTs compared methyldopa to all β-blockers grouped together and reported a combined RR of 1.19 (0.76, 1.84). Grouping labetalol together with other β-blockers is problematic because it has different receptor specificity and thus may have different effects on outcomes.

Labetalol binds to β-adrenergic receptors, lowering maternal heart rate and cardiac output, while also acting on α-adrenergic receptors in peripheral blood vessels to block the adrenergic stimulation that causes vasoconstriction. In contrast, methyldopa lowers blood pressure by binding to α2-adrenergic receptors as an agonist, reducing sympathetic outflow that causes peripheral vasoconstriction. Methyldopa crosses the placenta and recently, α2 receptors have been found on the placenta where they may regulate placental cell syncytialization and migration [40]. Methyldopa and labetalol may have differing effects on placental uptake of folate, a critical nutrient. Keating et al. found that labetalol exposure reduced the uptake of folate by placental cells and also decreased these cells' viability, while exposure to methyldopa did not [41]. Another mechanism through which antihypertensive medications could affect fetal growth is via methylation of placental DNA. Studies have found that alterations in placental DNA methylation are associated with maternal BP levels [42] and with infants being small for gestational age [43]. Placental genes that were affected included genes associated with cardio-metabolic disease [42] and with cell proliferation, protein transport, and inflammation [43]. We were not able to find studies that examined placental DNA methylation in relation to specific antihypertensive medications; this topic warrants further investigation.

We found a slightly higher risk of preterm delivery with nifedipine compared to labetalol in an analysis including over 4000 women. The Cochrane review found only one relevant RCT, a study of 112 women yielding an RR of 1.61 that was not statistically significant [44]. For NICU admission, we observed slightly higher risk with nifedipine than labetalol (aOR 1.21, 95% CI 1.02 to 1.43). Similarly, in a recent RCT, NICU admission was more frequent with nifedipine (18%) than labetalol (10%), yielding a risk difference of 7.8 (95% CI 2.2 to 13.4) [45]. The Cochrane meta-analysis reported a summary RR of 1.14 with a 95% CI of 0.63 to 2.05, which is wide enough to be consistent with our finding. Still, since our study was not randomized, our findings could reflect confounding, including by indication for use, since nifedipine is also used for tocolysis.

Current US guidelines recommend labetalol and nifedipine above other medications and state that methyldopa is less preferred because of possible lower effectiveness and adverse effects [6]. UK guidelines recommend labetalol, followed by nifedipine and then methyldopa [7]. There is little actual evidence to support this order of priority, and several older RCTs suggested that labetalol and methyldopa are equally effective in lowering BP [46–48]. While recognizing the potential for unmeasured confounding, our large observational study suggests that outcomes are very similar between methyldopa and labetalol, except for SGA. We suggest that when there is substantial concern for SGA, it may be reasonable to give more consideration to methyldopa.

This study has several strengths. The large population improves precision and allowed more granular analyses, including examining labetalol and nifedipine as individual agents and directly comparing antihypertensive medications. We studied a diverse population in community practice and adjusted for many covariates including BP. We had precise measures of gestational age and birthweight, not available in administrative (claims) datasets. We also had information about confounders not readily available in many large datasets, such as smoking, race/ethnicity and BMI. We conducted a validation study which demonstrated that our algorithm for preeclampsia had very high positive predictive value, and we validated potential stillbirths and terminations, reducing outcome misclassification.

The study also has limitations. There is potential for residual confounding because treatment was not randomized. Because we studied medication use in real world clinical practice,

there were not uniform criteria for initiating or intensifying antihypertensive medications. It is possible that women filled medications but did not take them, leading to misclassification of exposure. All women had health insurance and access to care and in general, their hypertension was well controlled at the time of the index fill, which may affect generalizability. Our data did not allow us to distinguish between spontaneous and indicated preterm birth, which on average would be expected to bias findings toward the null. The subgroup of women with gestational hypertension was too small to analyze separately. We did not have information about use of low dose aspirin, which the US Preventive Services Task Force recommended for women with chronic hypertension in 2014 [49]. The mean difference in birthweight between medications was small, and it could be argued that a difference this small is not clinically important. However, even a small shift of the birthweight curve to the left could result in a large relative increase in infants born SGA or with low birth weight, which may have important consequences for their long term health.

Our findings suggest a need for future research. We observed that labetalol appeared to convey higher risk of SGA. Infants born SGA may remain small, return to a normal growth curve, or experience compensatory weight gain leading to obesity, increasing future cardiometabolic risk. Future studies should examine child growth and development in relation to the use of specific antihypertensive medications during pregnancy. Other causal inference methods could be used to examine the associations we studied, including instrumental variable approaches such as Mendelian randomization. These observational analyses and designs rely on different untestable assumptions than the methods we utilized [50], and so if they found results similar to ours, this would further support a causal association.

In conclusion, in this large retrospective study, the prevalence of most maternal and infant outcomes was similar with different antihypertensive medications. A significantly lower risk of SGA was seen for methyldopa than labetalol, which is noteworthy because methyldopa is not preferred in US or UK guidelines [6, 7]. Our results suggest that methyldopa may warrant additional consideration, especially when there is heightened concern about growth restriction.

## Supporting information

**S1 File.**
(DOCX)

## Acknowledgments

### Prior presentation

Results were presented as an oral presentation at the International Conference on Pharmacoepidemiology, 35[th] annual meeting, in Philadelphia, Pennsylvania, from August 24–28, 2019 and additional results at the virtual International Conference on Pharmacoepidemiology, 36[th] annual meeting, September 16–17, 2020.

## Author Contributions

**Conceptualization:** Sascha Dublin, Lyndsay A. Avalos, T. Craig Cheetham, Thomas R. Easterling, Lu Chen, Victoria L. Holt, Romain S. Neugebauer, Sylvia E. Badon, Susan M. Shortreed.

**Data curation:** Nerissa Nance, Zoe Bider-Canfield.

**Formal analysis:** Abisola Idu, Susan M. Shortreed.

**Funding acquisition:** Sascha Dublin, Lyndsay A. Avalos, T. Craig Cheetham.

**Investigation:** Sascha Dublin, Lyndsay A. Avalos, T. Craig Cheetham.

**Methodology:** Romain S. Neugebauer, Susan M. Shortreed.

**Software:** Abisola Idu, Nerissa Nance, Zoe Bider-Canfield.

**Supervision:** Sascha Dublin, Lyndsay A. Avalos, T. Craig Cheetham, Romain S. Neugebauer, Kristi Reynolds, Susan M. Shortreed.

**Visualization:** Abisola Idu, Susan M. Shortreed.

**Writing – original draft:** Sascha Dublin, Abisola Idu, Susan M. Shortreed.

**Writing – review & editing:** Abisola Idu, Lyndsay A. Avalos, T. Craig Cheetham, Thomas R. Easterling, Lu Chen, Victoria L. Holt, Nerissa Nance, Zoe Bider-Canfield, Romain S. Neugebauer, Kristi Reynolds, Sylvia E. Badon, Susan M. Shortreed.

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
