## [Decision Letter · Decision Letter 0]

7 Oct 2021

PONE-D-21-28668Maternal and neonatal outcomes of antihypertensive treatment in pregnancy: A retrospective cohort studyPLOS ONE

Dear Dr. Dublin,

Thank you for submitting your manuscript to PLOS ONE. After careful consideration, we feel that it has merit but does not fully meet PLOS ONE’s publication criteria as it currently stands. Therefore, we invite you to submit a revised version of the manuscript that addresses the points raised during the review process.

We look forward to receiving your revised manuscript.

Kind regards,

Zhong-Cheng Luo

Academic Editor

PLOS ONE

Journal Requirements:

[I have read the journal's policy and the authors of this manuscript have the following competing interests:

SD received a grant to support this work from the National Institute on Child Health and Human Development. She has also received grant support from GSK for unrelated work.  

LC received a postdoctoral fellowship from the Group Health Foundation. She is now employed by Genentech (a member of Roche Group). 

ZBC is now employed by Roche Pharmaceuticals. 

TRE has consulted for Alnylam Pharmaceuticals, DiabetOmics, and Ferring Pharmaceuticals. 

SMS has received grant funding through her institutions from Syneos Health. 

LAA received funding through her institution from Bausch Health Companies and KR from Novartis and Merck & Co.]. 

4. We noted in your submission details that a portion of your manuscript may have been presented or published elsewhere. [Results were presented as an oral presentation at the International Conference on Pharmacoepidemiology, 35th annual meeting, in Philadelphia, Pennsylvania, from August 24-28, 2019 and additional results at the virtual International Conference on Pharmacoepidemiology, 36th annual meeting, September 16-17, 2020. ] Please clarify whether this conference proceeding was peer-reviewed and formally published. If this work was previously peer-reviewed and published, in the cover letter please provide the reason that this work does not constitute dual publication and should be included in the current manuscript.

Additional Editor Comments:

Please clarify the conflicts of interest with respect to the medications in the study, and whether any biases may arise.  

Please try to be more concise in Discussion.

Reviewers' comments:

Reviewer's Responses to Questions

**Comments to the Author**

1. Is the manuscript technically sound, and do the data support the conclusions?

Reviewer #1: Yes

Reviewer #2: Yes

2. Has the statistical analysis been performed appropriately and rigorously? 

Reviewer #1: Yes

Reviewer #2: I Don't Know

3. Have the authors made all data underlying the findings in their manuscript fully available?

Reviewer #1: Yes

Reviewer #2: Yes

4. Is the manuscript presented in an intelligible fashion and written in standard English?

Reviewer #1: No

Reviewer #2: Yes

5. Review Comments to the Author

Reviewer #1: The authors compared maternal and infant outcomes with different antihypertensive medications in pregnancy, and conclude that Risk of most outcomes was similar comparing labetalol, methyldopa and nifedipine, and SGA risk was substantially lower for methyldopa, suggesting this medication may warrant further consideration.

1. A retrospective cohort study with 6346 pregnant women is not enough to draw a robust conclusion, I am wondering if the authors may conduct a trans-ethnic meta-analysis based on similar published data with subgroup analyses according to age, ethnicity, geographical region, antihypertensive drug types. For this reason, the following papers can be cited and followed for the meta-analytic procedures (if the data is not enough available, at least DISCUSSION should be added as the LIMITATION of this study with enough citation to support the viewpoints):

Ref 1: Wu Y, et al. Multi-trait analysis for genome-wide association study of five psychiatric disorders. Transl Psychiatry. 2020 Jun 30;10(1):209.

Ref 2: Jiang L, et al. Sex-Specific Association of Circulating Ferritin Level and Risk of Type 2 Diabetes: A Dose-Response Meta-Analysis of Prospective Studies. J Clin Endocrinol Metab. 2019 Oct 1;104(10):4539-4551.

2. As is known, A retrospective cohort study is noty reliable compared to A prospectiv cohort study or Mendelian Randomization analysis that would be help for to disclose the causality.

But I strongly suggest to do causal inference analysis to see if the different antihypertensive treatment in pregnancy are causally triggering the different maternal and neonatal outcomes. If cannot, please discuss the limitations in the Discussion section in detail with additional citations to support the viewpoints. For these reasons, the following papers regarding causal inference in th Mendelian Randomization framework can be cited and followed.

Ref 1: Wang X, Fang X, Zheng W, Zhou J, Song Z, Xu M, Min J, Wang F: Genetic support of a causal relationship between iron status and type 2 diabetes: a Mendelian randomization study. J Clin Endocrinol Metab 2021.

Ref 2:Zhang, F. et al. Causal influences of neuroticism on mental health and cardiovascular disease. Hum. Genet. DOI: https://doi.org/10.1007/s00439-021-02288-x (2021).

Ref 3:Zhang, F. et al. Genetic evidence suggests posttraumatic stress disorder as a subtype of major depressive disorder. J. Clin. Investig. 27, 145942, DOI: https://doi.org/10.1172/jci145942 (2021).

Ref 4: Hou L, et al. Exploring the causal pathway from ischemic stroke to atrial fibrillation: a network Mendelian randomization study.Mol Med. 2020 Jan 15;26(1):7

3. In the Discussion section, the authors should discuss the potential mechanisms that is behand the conclusion, including antihypertensive drug targets and biological pathway, genetic and drug-induced epigenetic/epi-transcriptomic prenatal origins of small for gestational age (SGA), preterm delivery, preeclampsia, and stillbirth.

Reviewer #2: an excellent work that shed light on some of the effects of antihypertensive drugs used in pregnancy.

As the authors stated it is very difficult to get such information from randomized controlled presepective studies with such large numbers. This study help this gap in literature as well as it paves the way for other studies that may prospectively look at the effect of methydopa on SGA compared to other antihypertensive drugs.

Thank you and good luck.

6. PLOS authors have the option to publish the peer review history of their article (what does this mean?). If published, this will include your full peer review and any attached files.

Reviewer #1: No

Reviewer #2: **Yes: **Nourah Al Qahtani

---

## [Author Response · Author response to Decision Letter 0]

7 Apr 2022

Dublin et al.: Response to Reviewer Comments

Thank you for the opportunity to revise our manuscript. We appreciate the comments from the editor and reviewers and have sought to thoroughly address them. Please see our responses below. We first provide responses to the editor and reviewers’ substantive comments and at the bottom address additional journal requirements. All page and paragraph numbers below refer to the location in the “tracked changes” version of the manuscript.

Additional Editor Comments:

1. Please clarify the conflicts of interest with respect to the medications in the study, and whether any biases may arise. 

Response: We do not have any direct conflicts of interest. The sponsor of the study, the US National Institutes of Health, has no financial interest in the results. We did not receive funding for this study from any pharmaceutical company. As we reported, several authors have received research funding from pharmaceutical companies for topics unrelated to the current work. More specifically, none of the projects funded by these companies involved the medications studied in the current paper. Moreover, the medications being studied in the current paper are older medications that are no longer under patent, and so no single company is expected to benefit exclusively if one drug were to be found to be safer or more effective than the others. Thus, we do not believe that any biases will arise related to these potential interests we have reported. Below I will provide detail about specific funding received by different co-authors. 

Dr. Dublin has received research funding from GSK for unrelated work. GSK manufactures a combination medication used to treat hypertension in the general population, hydrochlorothiazide/triamterene. This drug is not recommended to treat pregnant women and was not examined in this study. 

Dr. Reynolds has received research funding from Amgen, Merck and Novartis. Amgen does not make medications used to treat hypertension. Both Merck and Novartis make antihypertensive drugs in the family of angiotensin converting enzymes inhibitors (ACEIs) or angiotensin receptor blockers (ARBs). It is widely recommended that these classes of medications not be used in pregnancy due to risk of birth defects. These drugs were not studied in our analysis. 

Dr. Easterling has received funding from Alnylam Pharmaceuticals and Ferring Pharmaceuticals. Neither company makes antihypertensive medications. He also has had funding from DiabetOmics, a company that makes point of care tests. One is a test for early detection of preeclampsia, which is an adverse outcome that can occur in women with hypertension in pregnancy. While this topic is tangentially related to our paper (we studied rates of preeclampsia with different medications), we do not feel there is any potential for a conflict. 

Dr. Chen currently works for Roche and Ms. Bider-Canfield for Genentech (a member of the Roche group of companies.) These companies do not make any medications to treat hypertension. These investigators did not work for Roche or Genentech at the time these analyses were conducted nor has their work at Roche or Genentech involved this subject matter. 

Dr. Avalos has had funding from Bausch Health Companies. Bausch makes several medications that are diuretics that can be used to treat hypertension. These include chlorothiazide, diltiazem, and ethacrynic acid. None of these drugs were studied in our paper. They are considered to be not favored in pregnancy according to current guidelines. 

Dr. Shortreed has received funding from Syneos Health for a study of safety of opioid medications. This study was required by the US Food & Drug Administration. Syneos Health was in charge of overseeing projects funded by the many different pharmaceutical companies that were mandated to fund studies on the safety of opioid medications. It is likely that some of these companies also make medications to treat hypertension. We are not aware of what medications each of these companies make that could be relevant to the current paper. Regardless, the research she participated in with funding from Syneos in had no overlap with the topic of the current paper. For the reasons discussed above, we do not believe this research funding has any potential for a conflict of interest regarding the current study. 

2. Please try to be more concise in Discussion.

Response: We have revised the Discussion to remove material and make it more concise. 

Reviewer #1: 

1. A retrospective cohort study with 6346 pregnant women is not enough to draw a robust conclusion, I am wondering if the authors may conduct a trans-ethnic meta-analysis based on similar published data with subgroup analyses according to age, ethnicity, geographical region, antihypertensive drug types. For this reason, the following papers can be cited and followed for the meta-analytic procedures (if the data is not enough available, at least DISCUSSION should be added as the LIMITATION of this study with enough citation to support the viewpoints):

Ref 1: Wu Y, et al. Multi-trait analysis for genome-wide association study of five psychiatric disorders. Transl Psychiatry. 2020 Jun 30;10(1):209.

Ref 2: Jiang L, et al. Sex-Specific Association of Circulating Ferritin Level and Risk of Type 2 Diabetes: A Dose-Response Meta-Analysis of Prospective Studies. J Clin Endocrinol Metab. 2019 Oct 1;104(10):4539-4551.

Response: We are unsure why the reviewer believes the current study is insufficient – whether the concern is that it is retrospective, or that the sample size is too small. We assume that the primary concern here is sample size, since that is a problem that a meta-analysis could overcome. 

The reviewer proposes that we conduct a trans-ethnic meta-analysis of “similar published data” including subgroup analyses according to many characteristics including age, ethnicity, geographic region, etc. We agree with the author that it would be a major benefit to the field if a meta-analysis could be conducted including a large sample size, drawing on data from rigorous randomized or observational studies, with detailed subgroup analyses. Unfortunately, the needed data do not yet exist, for several reasons. 

First, the body of literature from randomized trials is not sufficient to support a meta-analysis of this type. We have added new material in the introduction to make this clear. According to the most recent Cochrane review (2018)[1], a total of 29 studies have compared antihypertensive medications head to head for the treatment of hypertension in pregnancy; these studies included a total of 2774 women. These studies have been generally very small (including < 100 women, on average). Moreover, they have studied a large number of different medications. This means that for comparisons of specific medications or classes, the focus of this paper, often only 1 or 2 studies were available. In this situation, meta-analysis is not helpful. Our study population of 6346 women is over twice the size of the total number of women studied to date in randomized trials. This gave us larger numbers than the meta-analysis for class-specific comparisons, and we were able to compare individual medications in a more granular way. For instance, the Cochrane review made comparisons of all beta blockers vs. methyldopa for the outcome of small for gestational age, an outcome for which we had our most noteworthy finding. They identified a total of 6 RCTs including studies with a total of 577 women. They grouped all beta blockers together. We believe it is critical to study labetalol separately (because it has different receptor specificity and also is recommended as first line by current US guidelines, while other beta blockers are recommended not to be used in pregnancy). In our comparison of methyldopa vs. labetalol we were able to study 4851 women – more than 8 times as many as the Cochrane review. In summary, because of the dearth of randomized trials, and the small size and heterogeneity of existing trials, a meta-analysis of RCTs could not address the questions we examined. 

Perhaps the reviewer means that we should conduct a meta-analysis of prior observational studies. However, we note that the reviewer expresses concern about retrospective studies, which are the predominant type of study in the literature. Also, heterogeneity in these studies and potential for bias due to confounding would make it inappropriate to summarize them through meta-analysis. The prior studies focused on different medication exposures and used very different comparator groups. See the summary table below. Most studies compared a specific medication or class of medications to no exposure—meaning that most of the control group did not have hypertension. Because hypertension itself leads to adverse pregnancy outcome, comparing women treated for hypertension to healthy women is not helpful for understanding the risks and benefits of a medication, as this comparison is plagued by confounding by indication. Combining these studies into a meta-analysis would not advance the field. 

We agree that in the future, if more large and rigorous studies are published, such an effort would be very worthwhile. Unfortunately, at the current time the literature does not exist to support such a meta-analysis. 

Changes made to manuscript: We have added material to the introduction explaining why the literature is not sufficient to support a meta-analysis. We have added more material about the methodologic limitations of prior observational studies. See p. 4, paragraph 2, and p. 5, paragraphs 1-2. 

Table: Exposure and comparator groups in prior observational studies 

Author & year Exposure medication (N) Comparator group

Meidahl Petersen 2012[2]

All beta blockers (N=2459)

Labetalol (1452)

 Unexposed pregnant women

Orbach 2013[3]

Methyldopa (340), atenolol (107) Unexposed women; specifically excluded women with hypertension

Magee 2015[4]

Methyldopa (224) Labetalol (433)

Su 2013[5]

Many different classes were studied including beta blockers (414) and calcium channel blockers (303). Limited to women with chronic hypertension. Women without chronic hypertension

Xie 2014[6]

Labetalol (300) Methyldopa (923)

2. As is known, A retrospective cohort study is not reliable compared to a prospective cohort study or Mendelian Randomization analysis that would be help for to disclose the causality.

But I strongly suggest to do causal inference analysis to see if the different antihypertensive treatment in pregnancy are causally triggering the different maternal and neonatal outcomes. If cannot, please discuss the limitations in the Discussion section in detail with additional citations to support the viewpoints. For these reasons, the following papers regarding causal inference in the Mendelian Randomization framework can be cited and followed.

Ref 1: Wang X, Fang X, Zheng W, Zhou J, Song Z, Xu M, Min J, Wang F: Genetic support of a causal relationship between iron status and type 2 diabetes: a Mendelian randomization study. J Clin Endocrinol Metab 2021.

Ref 2:Zhang, F. et al. Causal influences of neuroticism on mental health and cardiovascular disease. Hum. Genet. DOI: https://doi.org/10.1007/s00439-021-02288-x (2021).

Ref 3:Zhang, F. et al. Genetic evidence suggests posttraumatic stress disorder as a subtype of major depressive disorder. J. Clin. Investig. 27, 145942, DOI: https://doi.org/10.1172/jci145942 (2021).

Ref 4: Hou L, et al. Exploring the causal pathway from ischemic stroke to atrial fibrillation: a network Mendelian randomization study.Mol Med. 2020 Jan 15;26(1):7

Response: We agree with the reviewer that it can be challenging to infer causality from observational study results because of potential for bias, including confounding. However, we disagree with the assumption that a prospective cohort study is inherently superior to a retrospective study. In our study, information about medication exposures came from computerized pharmacy data that are recorded prospectively, at the time of drug dispensing, well before any study outcomes occurred. Thus, using this approach to ascertain exposure avoids many of the limitations of other retrospective study designs (such as those that interview women after delivery about past exposures). Computerized pharmacy data are considered the “gold standard” for large scale pharmacoepidemiologic studies because they are prospectively recorded and shown to be more accurate than self-report or medication inventory. We have added a statement in the Methods about this (p. 7, paragraph 2). 

We agree with the reviewer that Mendelian randomization is an approach with potential to overcome some types of bias. There are multiple potential approaches to strengthen causal inference, of which Mendelian randomization is one, but not the only one. In designing this study, we selected methods that are considered rigorous causal inference methods by statisticians and epidemiologists.[7, 8]

First, we designed our study to emulate a target trial,[9] defining covariates before exposure and outcomes after exposure, something made possible by the electronic pharmacy data used to define exposure in our study. Second, we used inverse probability of treatment weighting to account for measured confounding, which is specifically designed to estimate the average treatment effect, that is, to provide a comparison of outcomes that would be expected if the whole population were to be treated with one medication versus another. Our target estimands are based on contrasts of counterfactual outcomes, which is the essence of causal inference.[10]

We agree that alternative approaches such as Mendelian randomization or another type of instrumental variable analysis could be used. And just as our study relies on the assumption of no unmeasured confounding, these alternative methods also rely on untestable assumptions.[11, 12] There is no one correct way to analyze data from observational studies, but a multitude of approaches each with their own set of assumptions. We considered the data available to us and several different possible approaches before deciding on the causal inference approach of inverse probability weighting. 

More specifically, responding to the request that we conduct an analysis using Mendelian randomization: We are not able to conduct such a study with the data we have available. We do not have access to genetic information for the over six thousand women in our cohort. In response to this review, we thought about how we would design such as study if genetic information on women were available. Designing such a study seems challenging, in particular because it seems possible that relevant mutations affecting genes involved in hypertension or its treatment (for instance, the receptors that beta blockers target) might prevent the development of hypertension in the first place, and so women with these mutations might simply not develop hypertension and might never require treatment in pregnancy. Our goal in this work was to help guide the choice of medications for women who do require treatment in pregnancy. Thus it seems possible that Mendelian randomization might not be suitable to address the specific clinical question we focused on. 

Changes made to manuscript: we have added material in the Introduction about the need for rigorous methods that can support causal inference (p. 5, paragraph 2, last sentence). In the Methods, we have added material in the Overview section stating that we attempted to emulate a target trial and chose methods that can support causal inference (p. 6, paragraph 1). We also added material to point out that the medication exposure data were recorded prospectively and thus not subject to recall bias (p. 7, paragraph 2). We added material in the Discussion saying that Mendelian randomization is a complementary approach that could be explored to strengthen causal inference (p. 20, paragraph 1). 

3. In the Discussion section, the authors should discuss the potential mechanisms that is behand the conclusion, including antihypertensive drug targets and biological pathway, genetic and drug-induced epigenetic/epi-transcriptomic prenatal origins of small for gestational age (SGA), preterm delivery, preeclampsia, and stillbirth.

Response: We appreciate the reviewer’s suggestion. Following this suggestion could lead to a very lengthy discussion because of the broad range of topics that the reviewer asks us to address. We have studied multiple drugs and multiple outcomes and the reviewer asks for discussion of multiple levels of mechanisms (genetic, epigenetic, epi-transcriptomic). This would be an interesting topic for a review article but is outside the scope of the current paper. In addition, the editor has asked us to make the Discussion more concise. 

We have added a paragraph in the Discussion focused on mechanisms by which labetalol and methyldopa might affect risk of SGA, because that is the outcome for which we have the most notable finding. See p. 17, paragraph 1. 

Reviewer #2: 

1. An excellent work that shed light on some of the effects of antihypertensive drugs used in pregnancy.

As the authors stated it is very difficult to get such information from randomized controlled presepective studies with such large numbers. This study help this gap in literature as well as it paves the way for other studies that may prospectively look at the effect of methydopa on SGA compared to other antihypertensive drugs.

Response: Thank you. We appreciate your interest in our work. 

Journal Requirements:

Response: Thank you, we have reviewed carefully and made changes if needed. 

Response: Thank you. We have confirmed that we are meeting those requirements. 

[I have read the journal's policy and the authors of this manuscript have the following competing interests:

SD received a grant to support this work from the National Institute on Child Health and Human Development. She has also received grant support from GSK for unrelated work. 

LC received a postdoctoral fellowship from the Group Health Foundation. She is now employed by Genentech (a member of Roche Group). 

ZBC is now employed by Roche Pharmaceuticals. 

TRE has consulted for Alnylam Pharmaceuticals, DiabetOmics, and Ferring Pharmaceuticals. 

SMS has received grant funding through her institutions from Syneos Health. 

LAA received funding through her institution from Bausch Health Companies and KR from Novartis and Merck & Co.]. 

Response: we have added the updated Competing Interests statement in the new cover letter. 

4. We noted in your submission details that a portion of your manuscript may have been presented or published elsewhere. [Results were presented as an oral presentation at the International Conference on Pharmacoepidemiology, 35th annual meeting, in Philadelphia, Pennsylvania, from August 24-28, 2019 and additional results at the virtual International Conference on Pharmacoepidemiology, 36th annual meeting, September 16-17, 2020. ] Please clarify whether this conference proceeding was peer-reviewed and formally published. If this work was previously peer-reviewed and published, in the cover letter please provide the reason that this work does not constitute dual publication and should be included in the current manuscript.

Response: We submitted an abstract to each conference, not a full manuscript. The abstracts were reviewed and rated by anonymous peer reviewers. The abstracts were published. Each abstract is very brief and does not contain anything approaching the depth of information contained in the manuscript. Thus, this work does not constitute dual publication. We have added a statement in the cover letter about this. 

Response: In the cover letter, we have described the restrictions on sharing a de-identified dataset and the process through which we can share data, and we provide contact information for the study data access committee, an institutional official, and an institutional review board. 

Response: We have reviewed the reference list and no changes were needed. No papers were retracted. 

 

REFERENCES

1. Abalos E, Duley L, Steyn DW, Gialdini C. Antihypertensive drug therapy for mild to moderate hypertension during pregnancy. The Cochrane database of systematic reviews. 2018;10(10):Cd002252.

2. Meidahl Petersen K, Jimenez-Solem E, Andersen JT, Petersen M, Brødbæk K, Køber L, et al. β-Blocker treatment during pregnancy and adverse pregnancy outcomes: a nationwide population-based cohort study. BMJ open. 2012;2(4).

3. Orbach H, Matok I, Gorodischer R, Sheiner E, Daniel S, Wiznitzer A, et al. Hypertension and antihypertensive drugs in pregnancy and perinatal outcomes. American journal of obstetrics and gynecology. 2013;208(4):301.e1-6.

4. Magee LA, von Dadelszen P, Singer J, Lee T, Rey E, Ross S, et al. Do labetalol and methyldopa have different effects on pregnancy outcome? Analysis of data from the Control of Hypertension In Pregnancy Study (CHIPS) trial. BJOG : an international journal of obstetrics and gynaecology. 2016;123(7):1143-51.

5. Su CY, Lin HC, Cheng HC, Yen AM, Chen YH, Kao S. Pregnancy outcomes of anti-hypertensives for women with chronic hypertension: a population-based study. PloS One. 2013;8(2):e53844.

6. Xie RH, Guo Y, Krewski D, Mattison D, Walker MC, Nerenberg K, et al. Association between labetalol use for hypertension in pregnancy and adverse infant outcomes. European journal of obstetrics, gynecology, and reproductive biology. 2014;175:124-8.

7. Hernán MA, Robins JM. Estimating causal effects from epidemiological data. Journal of epidemiology and community health. 2006;60(7):578-86.

8. Hernán MA, Robins JM. Causal Inference: What If. 1st ed. Boca Raton: Chapman & Hall/CRC; 2020.

9. Hernán MA, Robins JM. Using Big Data to Emulate a Target Trial When a Randomized Trial Is Not Available. American journal of epidemiology. 2016;183(8):758-64.

10. Pearl J. Causality: models, reasoning and inference. New York, N.Y.: Cambridge University Press; 2000.

11. Swanson SA, Tiemeier H, Ikram MA, Hernán MA. Nature as a Trialist?: Deconstructing the Analogy Between Mendelian Randomization and Randomized Trials. Epidemiology. 2017;28(5):653-9.

12. Hernán MA, Robins JM. Instruments for causal inference: an epidemiologist's dream? Epidemiology. 2006;17(4):360-72.

---

## [Editor Report · Decision Letter 1]

27 Apr 2022

Maternal and neonatal outcomes of antihypertensive treatment in pregnancy: A retrospective cohort study

PONE-D-21-28668R1

Dear Dr. Dublin,

We’re pleased to inform you that your manuscript has been judged scientifically suitable for publication and will be formally accepted for publication once it meets all outstanding technical requirements.

Kind regards,

Zhong-Cheng Luo

Academic Editor

PLOS ONE
---

## [Editor Report · Acceptance letter]

6 May 2022

PONE-D-21-28668R1 

Maternal and neonatal outcomes of antihypertensive treatment in pregnancy:
A retrospective cohort study 

Dear Dr. Dublin:

I'm pleased to inform you that your manuscript has been deemed suitable for publication in PLOS ONE. Congratulations! Your manuscript is now with our production department. 

Kind regards, 

on behalf of

Dr. Zhong-Cheng Luo 

Academic Editor

PLOS ONE